

# Modelling a Modern-like-pCO$_2$ Warm Period (MIS KM5c) with Two Versions of IPSL AOGCM

Ning Tan [1,2], Camille Contoux [2], Gilles Ramstein [2], Yong Sun [3], Christophe Dumas [2], Pierre Sepulchre [2]

[1]Key Laboratory of Cenozoic Geology and Environment, Institute of Geology and Geophysics, Chinese Academy of Sciences, Beijing 100029, China.

[2]Laboratoire des Sciences du Climat et de l'Environnement, LSCE/IPSL, CEA-CNRS-UVSQ, Université Paris-Saclay, F-91191 Gif-sur-Yvette, France.

[3]State Key Laboratory of Numerical Modelling for Atmospheric Sciences and Geophysical Fluid Dynamics, Institute of Atmospheric Physics, Chinese Academy of Sciences, Beijing, China

*Correspondence to: Ning Tan (ning.tan@mail.iggcas.ac.cn)*

**Abstract.** The mid-Piacenzian warm period (3.264 to 3.025 Ma) is the most recent geological period with a present-like atmospheric pCO$_2$ exhibiting significant warming relative to present conditions. With the advanced understanding of the climate variability of this interval, a specific interglacial (marine isotope stage KM5c, MIS KM5c, 3.205 Ma) is selected for Pliocene Model Intercomparison Project phase 2 (PlioMIP 2) and updated boundary conditions are provided. In this study, we carried out series of experiments according to the design of PlioMIP2 with two versions of the IPSL Atmosphere-Ocean Coupled General Circulation Model (AOGCM) (IPSL-CM5A and IPSL-CM5A2). By comparing with PlioMIP 1 experiment, run with IPSL-CM5A, our results show that the simulated MIS KM5c climate presents enhanced warming in mid-to-high latitudes, especially in ocean regions. This warming can be attributed to the largely enhanced Atlantic Meridional Overturning Circulation caused by the high latitude seaway changes. The tier experiments, conducted with IPSL-CM5A2 (with faster computation scheme), show that besides the increased pCO$_2$, both modified orography and reduced ice sheets contribute substantially in mid-to-high latitudes warming of MIS KM5c. When considering the pCO$_2$ uncertainties, the warming pattern changes, our model response to the variation of pCO$_2$ by +/-50ppmv is not symmetric in the surface air temperature, due to the non-linear response of the cryosphere (snow cover and sea ice extent). By analysing the Greenland Ice Sheet surface mass balance, we also demonstrate its vulnerability under both MIS KM5c and modern warm climate.

## 1 Introduction



The mid-Piacenzian warm period (MPWP; 3.264 to 3.025 Ma) is the most recent geological period with a present-like $pCO_2$ concentration and exhibiting significant warming relative to today. This interval has been intensively studied during the last three decades as it is generally considered to be a potential guide for future warming. There is abundance of marine and terrestrial data provided to represent the ocean/land temperatures, soil and vegetation conditions for this period. The

reconstructed $pCO_2$ data during MPWP range from 350 to 450ppmv (Bartoli et al., 2011; Pagani et al., 2010; Martínez-Botí et al., 2015), which are similar to present level. The MPWP is thought to be globally warmer by 2-4°C than preindustrial climate (e.g., Dowsett et al., 2009). A large warming amplification of 7-15°C is estimated in arctic regions derived from terrestrial proxies from the lake El'gygytgyn in NE arctic Russia (Brigham-Grette et al., 2013) and Ellesmere Island in North Arctic circle (Rybczynski et al., 2013). The meridional SST gradient is largely decreased due to the amplified warming in

the high latitudes. The zonal SST gradient is much weaker than present day as the ocean warm pool extends over most of the tropics (Brierley et al., 2009). The distribution of vegetation depicts a northward shift of boreal forest at the expense of tundra regions due to the warming conditions (Salzmann et al., 2008). Associated with this strong warmth, the reconstructed eustatic sea level is estimated to be 22(+/-10m) higher (between 2.7 and 3.2 Ma) than present (e.g., Miller et al., 2012) suggesting a large melting of Greenland ice sheet and a significant collapse of West Antarctic Ice sheet as well as unstable

regions of East Antarctic (Hill, 2009; Dolan et al., 2015; Koenig et al., 2015).

The initial purpose of studying on this period is to learn its relevance for the future climate change. However, considering the non-equilibrium state of the present climate due to the continuous change of forcing factors, the simulated stabilized MPWP may not be directly regarded as an analogue for future warming (Crowley, 1991). The importance of the MPWP

studies now is to investigate the abilities of climate models to produce warm climate and to study the respective contribution of forcing factors and feedbacks of internal climate components under warm conditions, which can also serve future climate projections. In Pliocene Model Intercomparison Project phase 1 (PlioMIP1), 11 models conducted the MPWP experiments. Among these results, there exists consistency in surface temperature change across models in the tropics and lack of consistency identified in model responses at high latitudes as well as total precipitation rate in the tropics (Haywood et al.,

2013). The modeled Atlantic Meridional Overturning Circulation and ocean heat transport for this interval in different models are likely unchanged relative to modern conditions (Zhang et al., 2013). However, when comparing to proxy data of sea surface and surface air temperature, climate models uniformly underestimate the warming in the high latitudes (Dowsett et al., 2012, 2013, Haywood et al., 2016b). Reasons for this discord between data and model are complex, but they can be attributed to three main aspects: boundary conditions uncertainty, modeling uncertainty (e.g., the model bias, annual

variability in the produced climatology fields) and data uncertainty (Haywood et al., 2013). In PlioMIP1, the MPWP is regarded simply as a stable interval despite of the climate variability existing over a 300-kyrs time slab due to the climate sensitivity and orbital parameters' change, thus the boundary conditions are made as an averaged condition over this long interval, whereas proxy data are representative of some orbital conditions inside this time slab. This boundary conditions uncertainty is thus considered as the main contributor to this data-model discrepancy (Haywood et al., 2016a). Therefore, the





ongoing PlioMIP phase 2 (PlioMIP2) changed strategy by choosing a representative interglacial during the MPWP interval: marine isotope stage KM5c (MIS KM5c; 3.205 Ma). Therefore, boundary conditions (known as PRISM4; Dowsett et al., 2016) have been updated for PlioMIP2, which include a new paleogeography reconstruction containing ocean bathymetry, and land/ice surface topography, which represent closure of Bering Strait and North Canadian Archipelago region and a reduced Greenland ice sheet by 50% in comparison to PlioMIP1. Besides, extra information of lake distribution and soil types (Pound et al., 2014) are also provided, but will not be used in this paper.

This study is conducted in the framework of PlioMIP2. Here we employ the new PRISM4 boundary conditions to conduct the MPWP experiments by using two versions of French AOGCM models (IPSL-CM5A and IPSL-CM5A2). The purpose of this study is to better understand the warm climate of the MPWP and to study the sensitivity of IPSL AOGCM model to the change of boundary conditions of land-sea mask and $pCO_2$. As IPSL AOGCM model has participated in PlioMIP1 (Contoux et al., 2012), we also compare the modelling results of PlioMIP2 with those of PlioMIP1 to quantify the impact of the high latitude seaways' changes on the climate system.

## 2 Model Descriptions

To accomplish the modelling work, we employed two versions of Institute Pierre-Simon Laplace (IPSL) coupled atmosphere-ocean general circulation model (AOGCM): IPSL-CM5A and IPSL-CM5A2. IPSL-CM5A is a low resolution coupled model which has been applied in CMIP5 for historical and future simulations (Dufresne et al., 2013) as well as for Quaternary and Pliocene paleoclimate studies (Kageyama et al., 2013; Contoux et al., 2012). IPSL-CM5A2 (Sepulchre et al., in prep) is an updated version of IPSL-CM5A. Critical changes from IPSL-CM5A include (i) technical developments to make IPSL-CM5A2 run faster (64yrs/day in CM5A2 instead of 8 years per day in CM5A), (ii) updates of the versions of components and (iii) a major tuning on the cloud radiative forcing to correct IPSL-CM5A cold bias in mid and high latitudes. Thus, to compare with PlioMIP1 (Contoux et al., 2012), we carried out PlioMIP2 core experiment with IPSL-CM5A, and conducted PlioMIP2 core experiment and tier experiments with IPSL-CM5A2 to save the computational cost. Components of these models are shortly presented as following. More details can be referred to Dufresne et al. (2013).

### 2.1 Atmosphere

The atmosphere component is LMDZ model (Hourdin et al., 2013) developed at Laboratoire de Météorologie Dynamique in France. This is a complex model that incorporates many processes decomposed into a dynamic part, calculating the numerical solutions of general equations of atmospheric dynamics, and a physical part, calculating the details of the climate in each grid point and containing parameterizations processes such as the effects of clouds, convection, orography (LMD_Modelling_Team, 2014). Atmosphere dynamics are represented by a finite-difference discretization of the primitive



equations of meteorology (e.g., Sadourny and Laval, 1984) on a longitude-latitude Arakawa C-grid (e.g., Kasahara, 1977). The chosen resolution of the model is 96x95x39, corresponding to an interval of 3.75 degrees in longitude and 1.9 degrees in latitude. There are 39 vertical levels, with around 15 levels above 20 km. This model has the specificity to be zoomed (the Z of LMDZ) if necessary on a specific region and then may be used for regional studies (e.g., Contoux et al., 2013). In IPSL-

CM5A2, re-tuning of the model has been done by altering the cloud radiative effect to decrease the cold bias of the model. More details can be found in Sepulchre et al (in prep).

## 2.2 Land

The land component in IPSL-CM5A and CM5A2 is ORCHIDEE (Organizing Carbon and Hydrology In Dynamic Ecosystems (Krinner et al., 2005) including three modules: hydrology, vegetation dynamics and carbon cycle. The hydrological module (Ducoudré et al., 1993) describes exchange of energy and water between atmosphere and biosphere, and the soil water budget (Krinner et al., 2005). Vegetation dynamics parameterization is derived from the dynamic global vegetation model LPJ (Sitch et al., 2003; Krinner et al., 2005). The carbon cycle model simulates phenology and carbon

dynamics of the terrestrial biosphere. Vegetation distributions are described using 13 plant functional types (PFTs) including agricultural C3 and C4 plants, which are not used in the MPWP simulations, bringing down the number of PFTs to 11, including bare soil. In this case, hydrology and carbon modules are activated, but vegetation is prescribed as the PlioMIP1 study by Contoux et al. (2012), using 11 PFTs, derived from the PRISM3 vegetation dataset (Salzmann et al., 2008). Therefore, soil, litter, and vegetation carbon pools (including leaf mass and thus LAI) are calculated as a function of

dynamic carbon allocation.

## 2.3 Ocean and sea ice

The ocean model included in IPSL-CM5A is NEMOv3.2 (Madec, 2008) which includes three principle modules: OPA (for
the dynamics of the ocean), PISCES (for ocean biochemistry), and LIM (for sea ice dynamics and thermodynamics). The configuration of this model is ORCA2.3 (Madec and Imbard, 1996), which uses a tri-polar global grid and its associated physics. The average horizontal resolution is 2° by 2°, refined at 0.5° in the tropics; vertical layers are 31. Temperature and salinity advection is calculated by a total variance dissipation scheme (Lévy et al., 2001; Cravatte et al., 2007). The mixed layer dynamics is parameterized using the Turbulent Kinetic Energy (TKE) closure scheme of Blanke and Delecluse (1993)

improved by Madec (2008). The sea ice module LIM2 is a two-level thermodynamic-dynamic sea ice model (Fichefet and Morales Maqueda 1997). Sensible heat storage and vertical heat conduction within snow and ice are determined by a three-layer model. OASIS model plays as a coupler (Valcke, 2006) to interpolate and exchange the variables and to synchronize the models. This coupling and interpolation procedures ensure local energy and water conservation. New version



NEMOv3.6 is included in IPSL_CM5A2 in which the river runoffs are now added through a non-zero depth, and have a specific temperature and salinity. The coupling system has been switched from OASIS3.3 to OASIS3-MCT (for Model Coupling Toolkit). More details are provided by Sepulchre et al (in prep).

## 3 Experiment Design

This section describes the boundary and the initial conditions imposed in our experiments. Here, the experiment names are generally consistent with the design of PlioMIP2 (Haywood et al., 2016a), they are referred by an abbreviated form E(x)(c), where c is the concentration of atmospheric $CO_2$ in ppmv and x represents boundary conditions that have been changed from the pre-industrial (PI) conditions, such that x can be absent for cases in which no boundary conditions have been modified or it can be "o" for a change in orography and/or "i" for a change in land ice configuration.

### 3.1 Pre-industrial experiments

The pre-industrial control simulation in IPSL-CM5A was performed as required by CMIP5/PMIP3 by the LSCE modelling group. It is a 2800-years simulation, which already started from equilibrium conditions. The pre-industrial control simulation in IPSL-CM5A2 was conducted by Sepulchre et al., (in prep) forced by CMIP5 pre-industrial boundary conditions and has 3000-years integration length.

### 3.2 Pliocene experiments

We have conducted six AOGCM experiments for the PlioMIP2 study, they are respectively core experiment Eoi400 with IPSL-CM5A model and core experiment Eoi400_v2 and four tier experiments E400_v2, Eoi450_v2, Eoi350_v2, Eo400_v2 with IPSL-CM5A2 model. As defined by the abbreviated form, the atmospheric $CO_2$ concentration imposed in each simulation can be referred to the number of the experiment's name (e.g., in the experiment "E400", the number "400" indicates that $pCO_2$ is set to 400 ppmv). Other greenhouse gases and orbital forcing are kept the same as IPSL PI control run (Table 1). Vegetation is kept the same as PlioMIP1 AOGCM simulation by Contoux et al. (2012). River routing and soil patterns are not changed in this study. Land sea mask in these experiments is modified from present, only by closing Bering Strait and North Canada Archipelago region, and modifying the topography in Hudson Bay (Figure 1). Ice sheet mask is referred to PRISM4 dataset (Dowsett et al., 2016) except for Eo400_v2 experiment, which is imposed with modern ice sheet. Topography in these five experiments are calculated based on modern topography used in IPSL model by superimposing on the anomaly between PRISM4 reconstructed topography and modern topography provided by PlioMIP2 database (Haywood et al., 2016a). When the new topography was lower than zero, absolute PRISM4 topography was implemented. Figure 1 shows the resulting topography in our PlioMIP2 experiments and topography anomaly between PlioMIP2 and PlioMIP1



experiments. The initial sea surface temperature and sea ice in Eoi400 and Eo400_v2 are derived from IPSL PlioMIP1 AOGCM simulation (Contoux et al., 2012). Eoi400 is conducted based on the equilibrium state of PlioMIP1 experiment (Contoux et al., 2012), with 650 years of integration length and integrated for 800 years, while Eoi400_v2 has 1500-years integration length. Average climatologies for these two experiments are calculated over the last 50 years. Four tier

experiments: E400_v2, Eoi450_v2, Eoi350_v2, Eo400_v2 are conducted based on the equilibrium state of Eoi400_v2 core experiment and have 400 years of integration length. Average climatologies for these four experiments are calculated over the last 30 years. Table 2 provides a summary for the experiments settings. Figure S1 shows time series of surface air temperature and deep ocean temperature at around 2.3km depth. For both core simulations, the trend in both global mean surface air temperatures ($< 0.18°C$ century$^{-1}$) and deep ocean temperature ($< 0.05°C$ century$^{-1}$) over the final 50 years of

integration are small. Other tier experiments also show relatively stable trends over the last 30 years of integration ($< 0.2°C$ century$^{-1}$ and $< 0.08°C$ century$^{-1}$ in surface air temperature and sub-surface ocean temperature respectively). Therefore, we consider model runs have reached a quasi-equilibrium state.

Although a standard $pCO_2$ of 400ppmv is selected for the Pliocene core experiments, the $pCO_2$ records during this interval

mostly range from 350 to 450ppmv. Thus, the tier experiments Eoi450_v2 and Eoi350_v2 are conducted to investigate the impact of $pCO_2$ uncertainty on the modelled Pliocene climate. The tier experiments E400_v2 and Eo400_v2 combined with core experiment Eoi400_v2 and PI control are used to quantify the relative importance of $pCO_2$, land ice and orography in the PlioMIP2 warmth. Because of the limitation of computational resources, we apply the linear decomposition for the forcing factors as: $dT_{CO2} = E400\_v2 - E280\_v2$ (1); $dT_{orography} = Eo400\_v2 - E400\_v2$ (2); $dT_{land\_ice} = Eoi400\_v2 -$

$E400\_v2$ (3)$\Delta T = dT_{CO2} + dT_{orography} + dT_{land\_ice}$ (4).

**4 Results and Discussion**

**4.1 Pliocene runs with IPSL-CM5A**

**4.1.1 Results in the Atmosphere**

Figure 2 shows the anomalies of global mean annual near surface air temperature (SAT, i.e. temperature at 2 meters), precipitation rate and sea surface temperature (SST) between PlioMIP experiments and pre-industrial control with IPSL-

CM5A. The global mean annual SAT in Eoi400 experiment is about 14.4°C which is 2.3°C warmer than that of pre-industrial. The warming in Northern Hemisphere (NH) high latitudes (>50°N) (4.2°C) is higher than that in the tropics (1.8°C). The magnitude of the warming for Eoi400 is slightly larger than that for PlioMIP1 experiment, which shows a global warming by 2.1°C. The major differences in SAT between Eoi400 and PlioMIP1 are found respectively in mid-

latitude Eurasia and arctic regions due to the change of regional topography and high latitude seaways as well as the reduced Greenland ice sheet. Thus, Eoi400 shows a reduced meridional temperature gradient than that in PlioMIP1 experiment. The global mean annual precipitation rate increases by 0.14 mm/d in Eoi400 due to the warming, the major increase locates in the monsoon regions and tropical oceans. The increased global mean precipitation rate as well as the monsoon area index

(Figure S2, calculated based on the method of Wang et al (2008)) in Eoi400 is similar to that in PlioMIP1. However, regional discrepancies still exist between these two experiments: the precipitation rates in Eoi400 in the tropics and NH high latitudes are higher than those in PlioMIP1 by 0.03 - 0.05 mm/d because of the increased warming in Eoi400 in these regions. Regional differences also exist in mountain regions (e.g., the Andes, the Rockies, Tibetan Plateau, the Himalayas and the Ethiopian Highlands) since these regions are modified largely in Eoi400 from the PlioMIP1 (Figure 1). In East Africa,

Eoi400 simulates an intensified precipitation than PlioMIP1, which is better consistent with proxy data from East Africa inferring a wet vegetation condition and hydrological systems during this period (Drapeau et al.,2014; Bonnefille 2010). Apart from the high latitude seaways' change, the regional difference in topography between PlioMIP2 and PlioMIP1 can also contribute to the rainfall change. Further sensitivity studies are needed to verify it.

**4.1.2 Results in the Ocean**

Accordingly, the global mean annual SST of Eoi400 is 1.7°C warmer compared to pre-industrial. It is 0.3°C warmer than PlioMIP1 and this warming majorly locates in the mid to high latitude oceans of the Northern Hemisphere. The warming in Eoi400 relative to PlioMIP1 can be attributed to the closure of Bering Strait and Canadian Archipelago, which is the major

difference in the boundary conditions between these two experiments. In the preindustrial control run (Figure 3a), the water flux through Bering Strait is about 1.0 Sv transporting much fresher and warmer water from the North Pacific to the Arctic Ocean. In Eoi400, as showed in Figure 3b, the water currents from the North Pacific to the Arctic through the Bering Strait and from the Arctic to the Baffin Bay are shut down. Consequently, the Arctic sea water gets much denser, then the wind-driven Beaufort gyre and transpolar drift get weakened (Figure 3c) and further reduce the associated East Greenland current

and the Labrador current, hence lead to saltier conditions in these adjacent regions (Figure 4b) resulting in the enhancement of the deep convection as well as the formation of North Atlantic Deep Water (Figure 4c, Figure 5b). The sea surface condition changes in North Atlantic region in Eoi400 (Figure 4) show agreement with the CCSM4 model results of Otto-Bliesner et al. (2017). Accordingly, we observe a strengthened Gulf Stream and North Atlantic currents as well as enhanced sub-polar gyre (Figure 3c), which can transport more heat to high latitudes (Figure 6b) and may link to a stronger convection.

Thus, a shallowed and enhanced AMOC by 4.9 Sv is observed in Eoi400, while AMOC in PlioMIP1 is not much different from modelled pre-industrial level (Figure 5). The increased AMOC resulting from the closure of Bering Strait and Archipelagos is likely consistent with previous studies of Hu et al. (2015), Kamae et al. (2016) and Chandan and Peltier (2017). However, the change of the AMOC strength in our PlioMIP2 simulation is much larger than other models. Hu et al. (2015) using CCSM3 and CCSM2 with different climate backgrounds show that the AMOC responses to the closure of the





Bering strait are about 2-3 Sv. Chandan and Peltier. (2016) show an increased AMOC strength by ~2 Sv after closing the Bering strait in the CCSM4 model. In the study of Kamae et al. (2016), with a different flux adjustment, they present a much stronger AMOC in their PlioMIP2 than their pre-industrial level. In fact, the simulated AMOC largely depends upon the vertical mixing schemes (Zhang et al., 2013). It is expected to see variations of simulated AMOC across models. Although

we observe a largely increased AMOC (15.7 Sv) in our PlioMIP2 simulation with IPSL-CM5A, the AMOC strength is still weaker than the modern observations (17.2 Sv, McCarthy et al.,2015). This is because the simulated modern AMOC (11 Sv) with this model is much weaker than the observations. Moreover, the simulated AMOC in PlioMIP1 with our model is also weaker than other models (Zhang et al.,2013). As shown in Figure 6, the total heat transport among PI control, Eoi400 and PlioMIP1 simulations is similar. The stronger AMOC in Eoi400 indeed strengthens the northward heat transport in the

Atlantic Ocean, while the weakened Pacific meridional ocean circulation in Eoi400 (PMOC, Figure S3), which contrasts with the data-based findings by Burls et al (2017), decrease the northward heat transport, thus leading to very slight change in total ocean heat transport.

The simulated warm conditions in high latitudes prevent sea ice from largely expanding during winter season and increase

sea ice melt during summer season (Figure 7). When compared to the PI condition, sea ice extent in the Eoi400 decreases by 5.4 Mkm$^2$ and 3.8 Mkm$^2$ respectively for the winter and summer season in the NH. In the SH, sea ice extent reduces by 8.8 Mkm$^2$ for the winter season and is nearly extinct during the summer. In comparison with PlioMIP1, NH sea ice cover in Eoi400 reduces by 2.1 Mkm$^2$ and 0.8 Mkm$^2$ respectively for cold and warm season but there is no large difference in SH between these two experiments. The largely decreased sea ice extent can amplify the warming in the high latitudes, through

its role as an insulation between the ocean and the atmosphere as well as positive albedo temperature feedback (Howell et al.,2014; Zheng et al., 2019). Reconstructed data in the Arctic Basin suggest the presence of seasonal rather than perennial sea ice in the Pliocene Arctic (Polyak et a., 2010; Moran et al.,2006), indicating a less or diminished summer sea ice cover. However, our IPSL model as well as half of participating models in PlioMIP1 cannot predict sea ice-free conditions during the summer season (Howell et al., 2016). Reasons for that are discussed in Howell et al (2016), which demonstrate the

unreasonable sea ice albedo parameterization for the warmer condition.

**4.2 Pliocene runs with IPSL-CM5A2**

**4.2.1 Results in the core experiment Eoi400_v2**

Figure 8 shows the anomalies of global mean annual near SAT (2-meter temperature), precipitation rate and SST between Eoi400_v2 and pre-industrial control with the identical model. The global mean SAT in Eoi400_v2 is about 15.3°C, which is 2.2°C warmer than pre-industrial conditions and the warming in high latitudes is much larger than in the tropics. It should be noted that the absolute SAT in Eoi400_v2 is warmer than that in Eoi400, while the SAT anomaly in Eoi400_v2 is weaker





than Eoi400. This is due to the cold bias correction between these two models: IPSL-CM5A2 presents a warmer pre-industrial condition by 1.1°C (Sepulchre et al., in prep) than that with IPSL-CM5A. The global mean annual precipitation rate increased by 0.13 mm/d in Eoi400_v2, which is comparable to the results of core experiments with IPSL-CM5A. In Eoi400_v2, the changes in the ocean conditions relative to its pre-industrial control are like Eoi400. The global mean annual

SST of Eoi400_v2 is 0.7°C warmer than Eoi400, the AMOC strength (Figure S4) in Eoi400_v2 is about 17.9 Sv which is 2.2 Sv larger than Eoi400, while AMOC anomaly is about 4.7 Sv relative to its pre-industrial level of 13.2 Sv, this anomaly is close to the result in IPSL-CM5A, indicating a coherent response of the AMOC to the same changes of boundary conditions. The sea ice cover is also largely decreased due to the warming in high latitudes (Figure 7).

**4.2.2 Relative importance of various boundary conditions in MIS KM5c warmth**

Figure 9 shows the relative contribution of various boundary conditions ($CO_2$ (a), Orography (b), Land ice (c)) on the warming of MIS KM5c calculated by the linear decomposition method.  Among these forcings, the increased $pCO_2$ by 120ppmv (from 280 to 400 ppmv) plays the most important role in both annual (+1.4°C) and seasonal warming (+1.38°C

and +1.48°C respectively in the summer and winter season). The changes of orography in PlioMIP2 also take an important control on annual mean warming (+0.51°C), especially for the north Atlantic and Barents Sea region. However, the changes in the orography decrease the temperature in the NH mid-to-high latitude inland regions, which may result from the change of North Pacific circulation. Seasonally, the orography changes contribute more importantly to the warming (+0.65°C) in summer than that in winter (+0.38°C). The decreased ice sheets impact is largely restricted to the high latitude regions and is

less important than the other two forcing factors in North polar region but plays the key role in the warming of South polar region. The mean annual warming resulting from the decreased ice sheets is about 0.25°C which is close to its summer and winter contribution. The residual impact besides the $pCO_2$, orography and land ice forcings is relatively small and negligible when making the linear decomposition of the forcing factors. This result shows some agreements with the study of Chandan and Peltier. (2018) in which they have applied non-linear decomposition.

**4.2.3 Greenland ice sheet instability under MIS KM5c warmth**

To understand how far could the Greenland ice sheet (GrIS) be sustained under a MPWP warm climate, we impose the modern GrIS into the Pliocene simulation (Eo400_v2). In comparison with PI control (Figure 10a), the mean annual surface

mass balance (SMB) in Greenland in Eo400_v2 (Figure 10b) is strongly negative around the coastal region, indicating vulnerable condition for costal ice sheet and ice shelves. This negative SMB condition majorly results from the increased summer temperature which leads to enhanced ablation around these regions (Figure S5). The mean annual SMB condition in Eo400_v2 is similar to that in modern condition (E400_v2, Figure10c). However, the warmer condition in Eo400_v2 bring





more precipitation in the South and Northwest Greenland, leading to enhanced accumulation (Figure S5), thus we observe increased SMB in these areas when compared to the PI control condition. In E400_v2, we also have increased SMB in these regions but much weaker than that in Eo400_v2, due to the different paleogeography settings as discussed above. Although these snapshot results cannot quantify the impact of the warm climate on the modern GrIS without considering the climate-ice sheet interaction, the results we get here can also herald the vulnerability of GrIS under such warm climate condition.

### 4.2.4 pCO$_2$ uncertainties in MIS KM5c warmth

Figure 11 depicts the anomalies of global mean annual SAT, precipitation rate and SST of Eoi450_v2 and Eoi350_v2 in

comparison with core experiment of Eoi400_v2. By increasing pCO$_2$ of 50ppmv in Eoi450_v2, global climate is slightly warmed up (+0.48°C) and the warming in high latitudes is more important (+0.7°C). However, when lowering pCO$_2$ by 50ppmv in Eoi350_v2, the change of climate is more important than that in Eoi450_v2, since we observe a global cooling of 0.71°C and cooling of 1.29°C over NH high latitudes. This asymmetric pattern in increasing/decreasing temperatures when augmenting/lowering pCO2 majorly results from the change of surface albedo associated with snow cover (not shown here).

In Eoi450_v2, the mean annual snowfall decreases by 6% between 40°N and 80°N when comparing to Eoi400_v2, while Eoi350_v2 represents an increased mean annual snow fall by 30% (not shown). The asymmetric pattern between Eoi450_v2 and Eoi350_v2 is also found in the changes of precipitation rates: Global climate gets slightly moister with an increased global precipitation rate by 0.02mm/d (+15%) in Eoi450_v2, while in Eoi350_v2, the global precipitation rate reduces by 0.04mm/d (-31%) and this reduction is more important in the tropical regions. Thus, our results can also show that the

response of IPSL coupled model to changing pCO$_2$ from 350 to 400 ppm is larger than from 400 to 450 ppm.

However, in the ocean, the increased or decreased SSTs resulting from augmenting or lowering pCO$_2$ of 50ppmv are likely symmetric. The AMOC strengths are also similar between Eoi450_v2 (17.4 Sv) and Eoi350_v2 (17.6 Sv)(Figure S4). Nevertheless, the changes of sea ice cover in these two experiments are unlike from each other (Figure 7). As in Eoi450_v2, the sea ice covers decrease slightly relative to Eoi400_v2 for both hemispheres (decreased by 0.2-0.5 Mkm$^2$ during cold

season and decreased by 0.01-0.2 Mkm$^2$ during warm season). Whereas in Eoi350_v2, the sea ice cover expands for both hemispheres, especially during the warm season in the NH (+1.7 Mkm$^2$).

### 4.3 Model-Data Comparison

Figure 12 shows the simulated mean annual SST anomalies (relative to PI experiments) of both core experiments (Eoi400, Eoi400_v2), together with the reconstructed SST (3.20 – 3.21Ma, Foley and Dowsett 2019) anomalies relative to near pre-industrial data (1870-1900, Rayner et al.,2003). The simulated SST anomalies in both core experiments are generally in phase with the reconstructed data. Some extremely warm sites are in disagreement with model results (e.g., Drilling sites in



North Greenland sea, in Mediterranean and Benguela current region). In summary, the simulated MIS KM5c SSTs generally underestimate the warming observed in the data, especially for the warming higher than 4°C (Figure 12b). Amongst these three experiments (PlioMIP1, Eoi400, Eoi400_v2), Both Eoi400 and Eoi400_v2 show increased warming in the mid-to-high latitudes when compared to the PlioMIP1 result. However, there is still an obvious discord in the strong warming between

5       model and data proxy, this may partly rely on the model performance. Moreover, the interpretation of the reconstructed data can also affect the data-model comparison results. Normally, SSTs were reconstructed from $U_{37}^{k'}$ paleothermometry assuming they represent annual mean values, whereas it has been shown that they can represent seasonal temperatures, for example representing the warmest summer month in the North Atlantic (NATL) (Leduc et al., 2017) and in the Benguela (Leduc et al., 2014). If we compare the reconstructed SST anomaly with the warmest summer month rather than the mean annual anomaly

from the model for the NATL and the Benguela region (Figure S6), the discrepancies between model and data will largely decrease. Moreover, in some regions, the modelling results overestimate the warming. To well understand the discord, more studies are further needed in the aspects of data interpretation as well as the multi-model comparison.

**5 Conclusions**

 In this paper, we describe the results of modelled warm interglacial of MIS KM5c (3.205 Ma) located in the interval of the MPWP (3.0-3.3 Ma) with imposing the new PRISM4 boundary conditions (Dowsett et al., 2016). Two versions of core experiments denoted Eoi400 and Eoi400_v2 are conducted based on two versions of IPSL coupled model: IPSL-CM5A and IPSL-CM5A2. Four tier experiments (E400_v2, Eoi450_v2, Eoi350_v2, Eo400_v2) are conducted with IPSL-CM5A2 to

study the relative contribution of each forcing factor in the warming climate. The new boundary conditions of PRISM4 adapted in our models produce an enhanced global warming in MIS KM5c, especially for the mid-to-high latitudes ocean when compared to the PlioMIP phase 1 results. The enhanced warming can be majorly attributed to the change of high latitude seaways which strengthens AMOC and transports more heat to high latitudes, and to the reduction of ice sheets and sea ice covers that largely decrease the outgoing shortwave radiation. The simulated warming conditions in MIS KM5c with

our models is weaker than other studies (e.g. Kamae et al., 2016; Chandan and Peltier, 2017). In our two core experiments, AMOC strengths increase remarkably (+4.7 Sv) in comparison with their related PI controls due to the closure of Bering Strait and North Canadian Archipelago regions. This result agrees with other studies (Kamae et al., 2016; Hu et al., 2015), but the extent of the increase of the AMOC highly depends upon the processes included in the ocean models.

Apart from the orography changes, the greenhouse gases emissivity and high latitude ice sheet configuration play also

important roles in the polar warm amplification, e.g., the reduced ice sheets in Antarctic play a key role in the Southern Hemisphere warming. The surface mass balance analysis show that the modern GrIS is vulnerable around the coastal regions under the warm MPWP as well as modern conditions. The model response to the $pCO_2$ uncertainties (+/-50ppmv based on the core simulation) is not symmetric in the surface air temperature, due to the non-linear response of the snow cover and sea ice extent. When the snow cover as well as sea ice has been largely decreased in area and duration, the sensitivity of climate



model to the growing $pCO_2$ may have a weaker thermal impact, in contrast to the near-linear manner of global surface air temperature responds to the cumulative emissions of $pCO_2$ in both the present short-term observations and transient modelling scenarios for the future. To conclude, further model inter-comparison work and data-model comparison work are needed to better understand the role of variable boundary conditions and the internal climatic processes in modelling the Pliocene warming climate.



**Figures and Tables**

**Table 1: Configuration common to all experiments described in this paper.**

| | |
|---|---|
| $CH_4$ | 760 ppb |
| $N_2O$ | 270 ppb |
| $O_3$ | Local modern |
| $CFC_s$ | 0 |
| Solar constant | 1365 $W/m^2$ |
| Eccentricity | 0.016715 |
| Obliquity | 23.441 |
| Perihelion | 102.7 |
| Dynamic vegetation | Off |
| Soil types and lakes | Local modern |

**Table 2: Details of experimental settings.**

| Exp names | Models | Topography &Ice sheet | $CO_2$ (ppmv) | Integration length (yrs) | Climatologies |
|---|---|---|---|---|---|
| Eoi400 | IPSL-CM5A | PRISM4 | 400 | 650+800 | Last 50 yrs |
| Eoi400_v2 | IPSL-CM5A2 | PRISM4 | 400 | 1500 | Last 50 yrs |
| Eoi450_v2 | IPSL-CM5A2 | PRISM4 | 450 | 1500+400 | Last 30 yrs |
| Eoi350_v2 | IPSL-CM5A2 | PRISM4 | 350 | 1500+400 | Last 30 yrs |
| Eo400_v2 | IPSL-CM5A2 | Modern Ice sheet, PRISM4 topo in other regions | 400 | 1500+400 | Last 30 yrs |
| E400_v2 | IPSL-CM5A2 | Modern | 400 | 1500+400 | Last 30 yrs |



**Table 3: Diagnostics for each experiment. Anomalies are calculated by comparing the related PI controls.**

| Exp names | MA SAT & PRECIP (Anomaly) (units:℃ & mm/d) | | | MA SST (Anomaly) (unit: ℃) | AMOCindex (unit: Sv) |
|---|---|---|---|---|---|
| | Global | Tropics | High Latitudes (NH) | | |
| PlioMIP 1 | 2.1 & 0.13 | 1.7 & 0.17 | 3.9 & 0.21 | 1.4 | 10.8 |
| Eoi400 | 2.3 & 0.14 | 1.8 & 0.20 | 4.2 & 0.28 | 1.7 | 15.7 |
| Eoi400_v2 | 2.2 & 0.13 | 1.6 & 0.19 | 3.8 & 0.23 | 1.6 | 17.9 |
| Eoi450_v2 | 2.6 & 0.15 | 2.1 & 0.23 | 4.5 & 0.27 | 1.9 | 17.4 |
| Eoi350_v2 | 1.5 & 0.09 | 1.0 & 0.13 | 2.5 & 0.14 | 1.2 | 17.6 |
| Eo400_v2 | 1.92 & 0.12 | 1.56 & 0.18 | 3.56 & 0.23 | 1.5 | 17.4 |

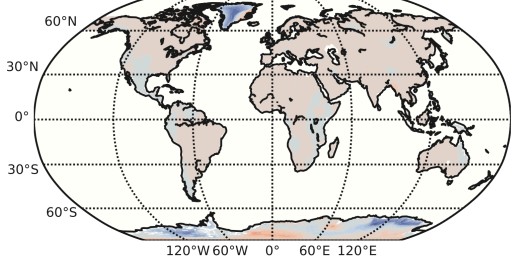

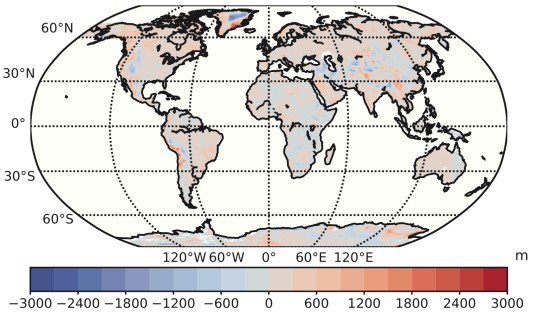

**Figure 1: Anomalies prescribed in topography of PlioMIP2 respectively in relative to PI control (upper) and PlioMIP 1 (lower).**

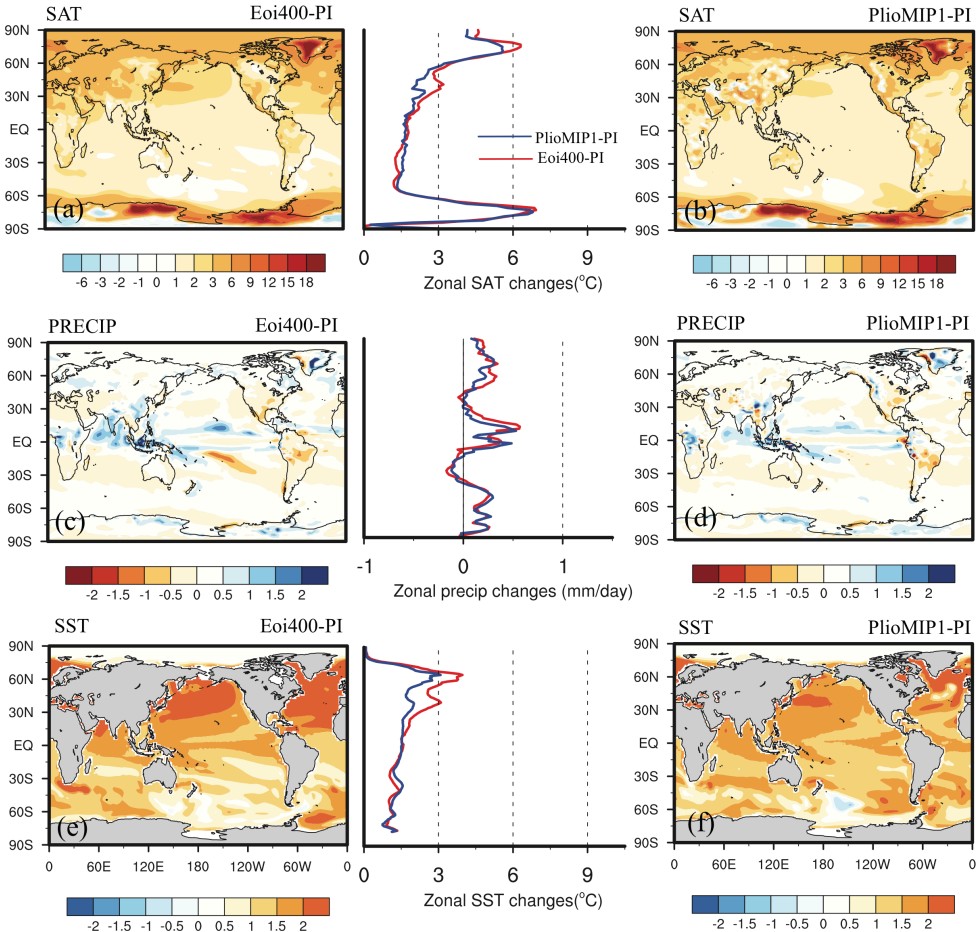

**Figure 2: Anomalies of mean annual SAT (a, b), mean annual precipitation rates (c, d) and mean annual SST for PlioMIP 2 (Eoi400) and PlioMIP 1 conducted with IPSL-CM5A in comparison with associated pre-industrial control experiment. The middle panel represents the zonal mean of related anomalies (red lines for Eoi400, blue lines for PlioMIP 1).**



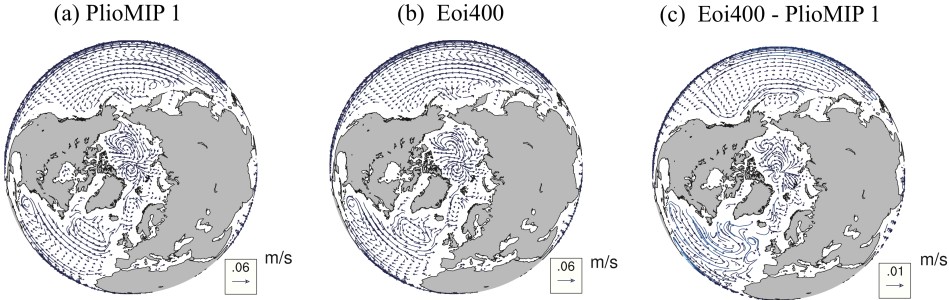

**Figure 3: Mean annual Ocean current above 500 meters for PlioMIP 1 (a) and Eoi400 (b), (c) shows the difference in ocean current between Eoi400 and PlioMIP1.**

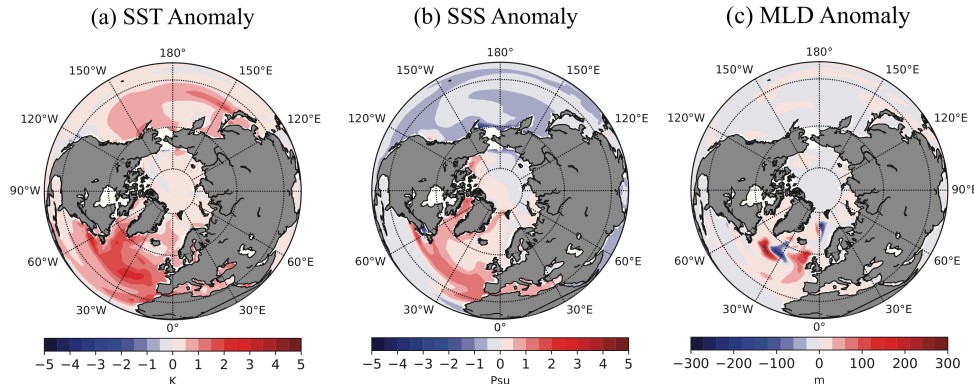

**Figure 4: The differences in the mean annual sea surface temperature (a), sea surface salinity (b) and the mixed layer depth (c) between Eoi400 and PlioMIP 1 experiment.**



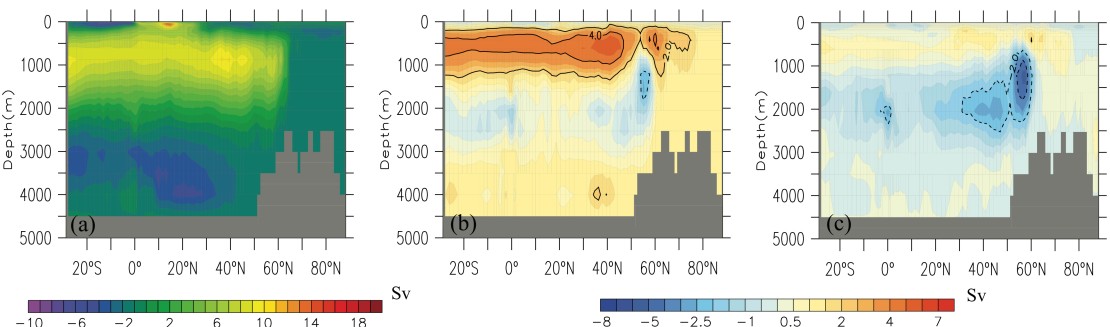

**Figure 5: Mean annual AMOC of PI control (a) and AMOC anomalies of Eoi400 (b) and PlioMIP 1 (b) in comparison with PI condition.**

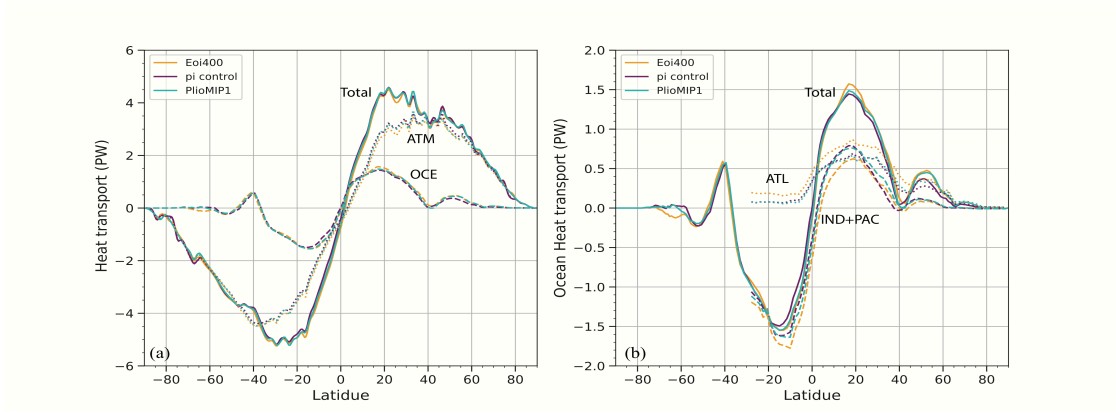

**Figure 6: Meridional heat transport in both atmosphere and ocean (a), Meridional ocean heat transport in different regions (b). (Orange, purple and blue lines represent respectively for the results of Eoi400, PI control and PlioMIP 1)**



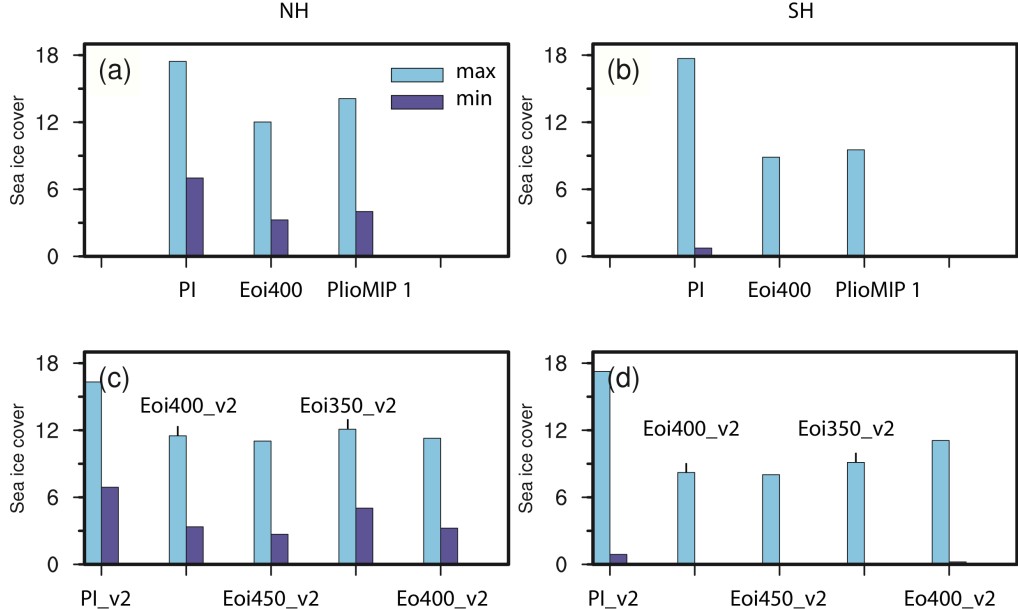

**Figure 7: Maximum and Minimum sea ice covers for both hemispheres in each experiment (unit:1E+106 km2).**



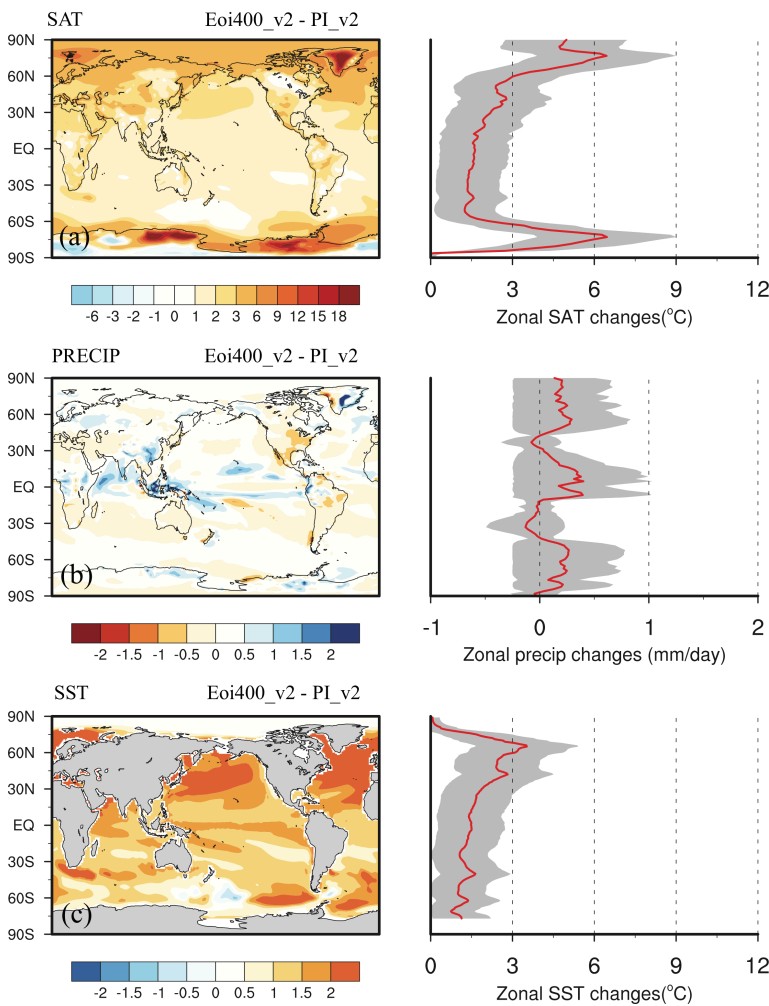

**Figure 8: Anomalies of mean annual SAT(a), mean annual precipitation (b) and mean annual SST (c) of Eoi400_v2 in comparison with associated PI control experiment. The right panel represents the zonal mean of related anomalies; the shaded area shows the one sigma standard deviation.**

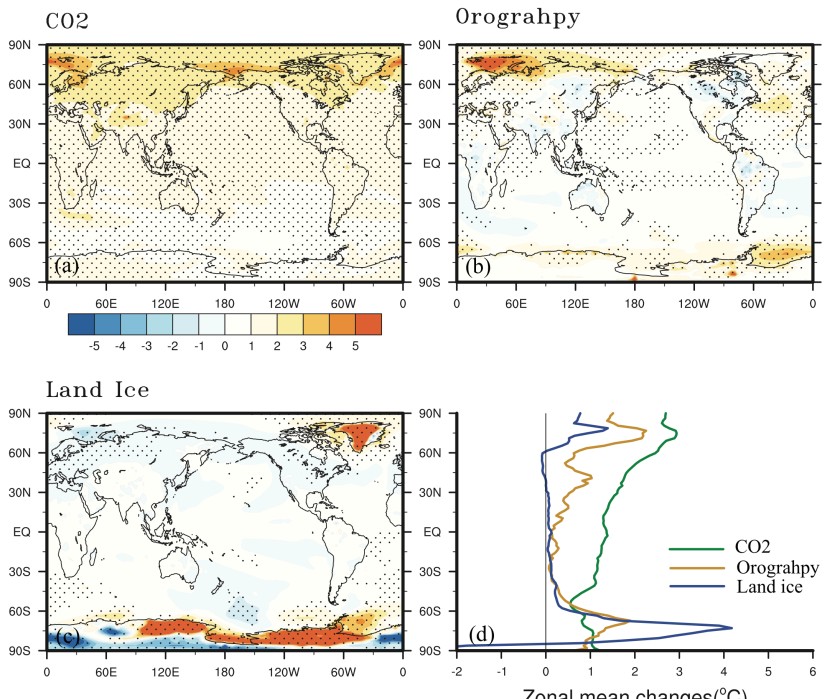

**Figure 9: The relative contribution of various boundary conditions (CO2 (a), Orography (b), Land ice (c)) on the warmth of PlioMIP 2 and their zonal mean values (d). Stippling indicates regions where results are statistically significant at a 99% confidence criteria.**





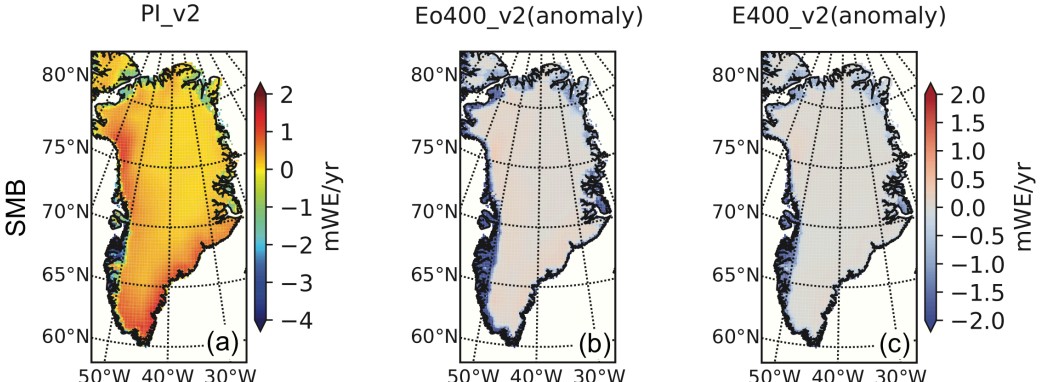

**Figure 10: Mean annual surface mass balance (SMB) in Greenland in PI control experiment (a) and the anomalies of SMB in Eo400_v2 and E400_v2 experiments in comparison with PI control (unit: mWE(water equivalent)/yr).**



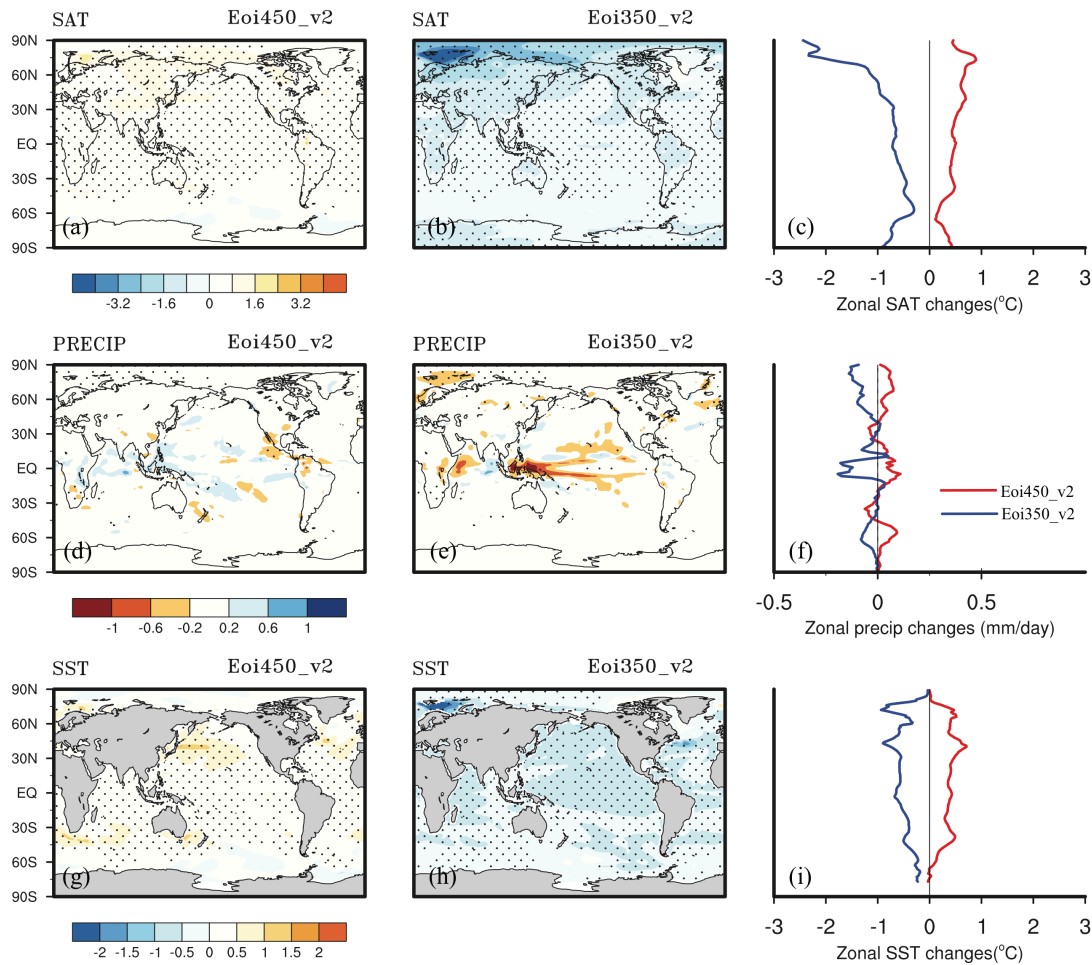

**Figure 11: Anomalies of mean annual SAT, mean annual precipitation rate and mean annual SST for Eoi450_v2(a, c, e), Eoi350_v2(b, d, f) in comparison with Eoi400_v2. The last column (c, f, j) of this panel shows the zonal mean of related anomalies (red and blue lines represent respectively for the results of Eoi450_v2 and Eoi350_v2). Stippling indicates regions where results are statistically significant at a 99% confidence criteria.**





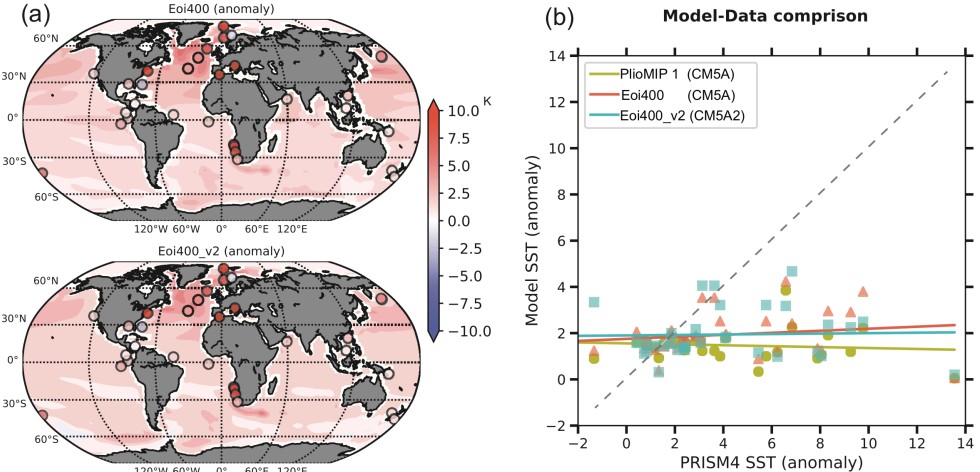

**Figure 12: SST model data comparison. (a) Modelled mean annual SST anomalies of MIS KM5c (in relative to PI controls) and reconstructed MIS KM5c SST anomalies (in relative to near pre-industrial data). (b) The relationship between modelled SST**
**anomalies and PRISM4 data anomalies.**

**Data availability**: Climatological averages of each simulation in NetCDF format will be uploaded to the PlioMIP2 data repository soon (sftp://see-gw-01.leeds.ac.uk). Specific data requests should be sent to the lead author (ning.tan@mails.iggcas.ac.cn). All PlioMIP2 boundary conditions are available on the USGS PlioMIP2 web page (http :
//geology.er.usgs.gov/egpsc/prism/7_pliomip2/).

**Author Contributions**: N. T., G. R. and C. C. designed the study. N. T. conducted the model set-up, spin-up and major data analysis and wrote the manuscript. Y. S., C. C. and C. D. contributed to discuss the data analysis and the structure of this work. P. S. provided the IPSL-CM5A2 information and its related control run simulation. All co-authors helped to improve
this manuscript. Correspondence and requests for materials should be addressed to N. T.

**Competing interests**: The authors declare that they have no conflict of interest.

**Acknowledgements**: We thank Oliver Marti and Jean-Baptiste Ladant for their helps on the model set-ups. This study was
performed using HPC resources from GENCI-TGCC under the allocation 2019-A0050102212 and supported by French State Program Investissements d'Avenir (managed by ANR), ANR HADOC project (ANR-17-CE31-0010) , by the



"Strategic Priority Research Program" of the Chinese Academy of Sciences (Grant No.XDB 26000000) and by the Major Program of National Science Foundation of China (Grant No. 41690114)

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
