# Peer review of "Modelling a Modern-like-pCO2 Warm Period (MIS KM5c) with Two Versions of IPSL AOGCM"

_Climate of the Past, 2019_

## Referee Comment (RC1) · Anonymous Referee #1 · 5 Aug 2019

In the paper, the authors document their Pliocene experiments with IPSL-CM5A and CM5A2, which contribute to the PlioMIP2. They carry out the experiments following the PlioMIP2 guideline. The results are clearly presented. I suggest that the paper should be accepted after minor revisions.

I have two suggestions that authors should consider in the revised version.

1. The energy balance at the top of atmosphere should be added in Table 3.

2. In this study, the authors use modern river routing, but modify the land-sea mask (closing the Bering Strait, the Canadian Archipelago and the Hudson Bay). In this way, rivers might not reach ocean, in particular in the Hudson Bay. It is likely that the simulated responses in AMOC are caused not only by the closing of these seaways, but

also the salinity drift in the Pliocene simulations. To exclude this possibility, I suggest the authors check the mean salinity in ocean, and add them in Table 3.

Other minor corrections

Page 3 line 5, "therefore" also appears in the previous sentence. Reword.

Page 3 line 6, "but will not be use in this paper" change to "but not used in this study".

Page 4 line 3, 20km or 2km?

Page 4 line 19, please check if "litter" is rightly used in the sentence.

Page 5 line 21, respectively should be deleted.

Page 5 line 22, the second "and" can be changed to as well as

Page 5 line 28, "only by closing Bering Strait and North Canada Archipelago region, and modifying the topography in Hudson Bay", chang to "only by closing the Bering Strait and the North Canadian Archipelago region, and modifying the topography in the Hudson Bay"

Page 7 line 21, add the before Bering Strait

Page 7 line 32, add Canadian

Page 9 line 11, "contribution" to "contributions"

---

## Referee Comment (RC2) · Anonymous Referee #2 · 6 Sep 2019

The comment was uploaded in the form of a supplement:
https://www.clim-past-discuss.net/cp-2019-83/cp-2019-83-RC2-supplement.pdf

2

---

## Author Comment (AC1) · 23 Sep 2019

Anonymous Referee #1 In the paper, the authors document their Pliocene experiments with IPSL-CM5A and CM5A2, which contribute to the PlioMIP2. They carry out the experiments following the PlioMIP2 guideline. The results are clearly presented. I suggest that the paper should be accepted after minor revisions. I have two suggestions that authors should consider in the revised version.

Response: Thanks a lot for these positive comments on this paper.

1. The energy balance at the top of atmosphere should be added in Table 3.

Response: We have added this in Table 3.

[Figure]

2. In this study, the authors use modern river routing, but modify the land-sea mask (closing the Bering Strait, the Canadian Archipelago and the Hudson Bay). In this way, rivers might not reach ocean, in particular in the Hudson Bay. It is likely that the simulated responses in AMOC are caused not only by the closing of these seaways, but also the salinity drift in the Pliocene simulations. To exclude this possibility, I suggest the authors check the mean salinity in ocean, and add them in Table 3.

Response: Thanks for this suggestion. The river routing is generally kept the same as the PI configuration. But if the estuaries change to land after the land-sea mask modification, the new estuaries are created towards the closest ocean. So, we confirm river water have flowed into the ocean and there is no salinity drift in our experiments. As you suggested, we have added the mean ocean surface salinity in Table 3.

Other minor corrections .

1. Page 3 line 5, "therefore" also appears in the previous sentence. Reword.

Response: Corrected.

2. Page 3 line 6, "but will not be use in this paper" change to "but not used in this study". Page 4 line 3, 20km or 2km?

Response: Corrected. It is 20 km. In LMDZ, there are 39 levels in vertical in total and above 20 km, there are 15 levels.

3. Page 4 line 19, please check if "litter" is rightly used in the sentence.

Response: We change to "litterfall" which is better.

4. Page 5 line 21, respectively should be deleted.

Response: Corrected.

5. Page 5 line 22, the second "and" can be changed to as well as

Response: Changed as suggested.

6. Page 5 line 28, "only by closing Bering Strait and North Canada Archipelago region, and modifying the topography in Hudson Bay", change to "only by closing the Bering Strait and the North Canadian Archipelago region, and modifying the topography in the Hudson Bay"

Response: Changed as suggested.

7. Page 7 line 21, add the before Bering Strait

Response: Corrected.

8. Page 7 line 32, add Canadian Page 9 line 11, "contribution" to "contributions"

Response: Corrected.

---

## Author Comment (AC2) · 23 Sep 2019

Anonymous Referee #2 I would like to begin by apologizing to the authors for the considerable delay in getting my review back to them. I was occupied throughout August on a personal matter and my expectation of still being able to review the paper on time proved over-optimistic. Nevertheless, I should have known better and I apologize for my failing once again. Overall I have found the paper to be scientifically sound. The only principle issue with the manuscript in the current form is the language which requires considerable editing. With this review I am attaching a copy of the paper that has been extensively annotated using Acrobat Reader for language and technical edits. The following document therefore only contains points not included on the annotated manuscript. The paper can be accepted with minor changes.

[Figure]

Response: Thanks a lot for the detailed language editing and constructive comments which largely improve the quality of this paper. Concerning the editing and comments in the PDF, we have corrected directly in the revised paper and these comments are not listed here.

1. The abstract sentence spanning lines 23-25 is not very clear and also potentially confusing. Please consider re-writing this.

Response: We have re-written this sentence as "When considering the pCO2 uncertainties (+/-50 ppmv) during the Pliocene, the responses of the modelled mean annual surface air temperature to changes to pCO2 (+/-50 ppmv) are not symmetric, which is likely due to the non-linear response of the cryosphere (snow cover and sea ice extent)." (Revised version Page 1, lines 23 -25)

2. On Page 2 regarding the comment about the zonal SST gradient in line 10: I think there is still considerable discussion in the literature about this aspect of the Pliocene climate. So I don't think the sentence should only provide one point of view.

Response: Thanks for this suggestion. We have re-organized this sentence to include more point of view on the reduced zonal SST gradient: "The zonal SST gradient is much weaker than present day (Wara et al., 2005; Ravelo et al., 2006; Fedorov et al., 2013). Different causes have been investigated for this weaker zonal SST gradient during the Pliocene. Brierly et al (2009) argue that the ocean warm pool expansion over most of the tropics can be responsible for the reduced zonal SST gradient (Brierley et al., 2009). Some researchers argue that a reduction in the meridional gradient of cloud albedo can sustain the reduced zonal and meridional SST gradient (Burls and Fedorov 2014)." (Revised version, Page 2, lines 10-14.)

3. On Page 5 after the end of paragraph on section 3, please add: Because we report on experiments performed with two versions of the IPSL model, we indicate the experiment conducted using the updated version of the model by the suffix "_v2".

Response: We have added this sentence in the related place.

4. Page 5 line 16: what equilibrium conditions did the model start from?

Response: Here, the model starts from the conditions that the carbon pools are close to equilibrium in the coupled model. More details are provided in Dufresne et al., 2013, section 4.1. For your convenience, we copy the paragraph here: "The initial state of the IPSL-CM5A-LR model was obtained in four steps. First, a 2,500-year long simulation of the oceanic model without carbon cycle where the atmospheric conditions are imposed and correspond to the version 2 of the Coordinated Ocean-ice Reference Experiments data sets (Large and Yeager 2009) was achieved. Second, the full carbon-cycle configuration of the IPSL- CM5A-LR model was integrated for a period of 600 years with the solar constant and the concentrations of GHGs and aerosols corresponding to their pre-industrial values. Third, because this last simulation is too short for the ocean and biosphere carbon pools to reach equilibrium, offline simulations a few thousand year-long with the ocean and land carbon cycle models (ORCHIDEE and PISCES) were conducted separately. These offline simulations were forced by the atmospheric and oceanic variables from the preceding 600-year simulation and by a constant preindustrial value for the atmospheric CO2. Fourth, and once the carbon pools are equilibrated, their values are included back into the complete IPSL-CM5A-LR model, which is again integrated for another 400 years. At this time, carbon pools are close to equilibrium in the coupled model as well. This long integration is used as initial state for the control pre-industrial simulations."

5. Page 5: The statement "River routing and soil…." You mean unchanged from PI?

Response: Sorry, we need to clarify the details here. The soil types and river routing are generally kept the same configuration with PI, except the regions where the changes to the topography modify the river routing and the estuaries. We have added this clarification in the related place. (Revised version, Page 5, lines 32-34.)

6. Page 6, line 1: You mean Eoi400_v2 and not Eo400_v2 right?

Response: It was a typo. It is Eoi400_v2. We have corrected it.

7. Page 6, lines 2-3 I don't follow the meaning of the sentence on these lines up to the "800 years".

Response: We have re-written the sentence as "Eoi400 has run for 800 modelling years and the initial condition is from the equilibrium state of PlioMIP1 experiment (Contoux et al., 2012), which has 650-years integration length. Eoi400_v2 has run for 1500 modelling years." (Revised version, Page 6, lines 8-10.)

8. Page 7. Please break and re-write the long sentence spanning 23-26. There is too much there.

Response: We have re-written this sentence as "Consequently, the Arctic sea water gets much denser and thus the wind-driven Beaufort gyre and transpolar drift get weakened (Figure 3c). The associated East Greenland current and the Labrador current get weaker resulting in saltier conditions in these adjacent regions (Figure 4b). Thus, the deep convection and the formation of North Atlantic Deep Water (Figure 4c, Figure 5b) over these regions enhance." (Revised version, Page 7, lines 31-33 and Page 8, line 1)

9. Page 10: Foley and Dowsett 2019 reference is not present in the list of references. Whereas the list contains a Dowsett 2019 reference that seems incorrect.

Response: Corrected.

10. Page 13, Table 2: Please also put in details of the PI experiment to this table

Response: We have put the PI information in Table 2.

11. Page 14, Table 3 new suggested title: Diagnostics for each experiment. The anomalies are computed against the PI controls corresponding to the version of the numerical model employed"

Response: Changed as suggested.

[Figure]

12. Page 14, Fig 1 suggested title: Anomalies of the PlioMIP2 topography relative to PI control (upper) and PlioMIP 1 (lower).

Response: Changed as suggested.

13. Page 21, I suggest a different color scheme for sub-figures (b) and (c) and the inclusion of a 0 value contour line.

Response: Figure 10 is modified as suggested.

———————————————————